# T Cell/B Cell Interactions in the Establishment of Protective Immunity

**DOI:** 10.3390/vaccines9101074

**Published:** 2021-09-24

**Authors:** Julia Ritzau-Jost, Andreas Hutloff

**Affiliations:** Institute of Immunology, University Hospital Schleswig-Holstein, 24105 Kiel, Germany; julia.ritzaujost@gmail.com

**Keywords:** follicular helper T cells, peripheral helper T cells, germinal center reaction, vaccination, inflamed tissues, cytokines

## Abstract

Follicular helper T cells (Tfh) are the T cell subset providing help to B cells for the generation of high-affinity antibodies and are therefore of key interest for the development of vaccination strategies against infectious diseases. In this review, we will discuss how the generation of Tfh cells and their interaction with B cells in secondary lymphoid organs can be optimized for therapeutic purposes. We will summarize different T cell subsets including Tfh-like peripheral helper T cells (Tph) capable of providing B cell help. In particular, we will highlight the novel concept of T cell/B cell interaction in non-lymphoid tissues as an important element for the generation of protective antibodies directly at the site of pathogen invasion.

## 1. Introduction

Vaccination strategies against infectious diseases are typically based on the generation of long-lived high-affinity antibodies, which are capable of combating pathogens in multiple mechanisms. This includes neutralization of toxins or prevention of receptor-specific virus entry into host cells. Moreover, antibody opsonization of pathogens constructively promotes their destruction mediated by the complement system or phagocytosis [1]. Despite the immense success of vaccinations, which has effectively eliminated or vastly reduced a wide range of formerly devastating infections, there still remains a wide range of communicable diseases for which efficient vaccination strategies are in order. Furthermore, many of the existing vaccine regimens only provide partial immunity or require booster immunizations at rather short intervals. To improve vaccination regimes as well as to develop novel strategies, it is vital to comprehend the process of protective antibody generation in humans.

Antibodies are produced by plasma cells, which reside in long-term survival niches such as the bone marrow, the spleen and mucosal tissues, where they ideally provide protective and lifelong levels of antibodies [2]. Concurrently, a second line of defence is formed by memory B cells, which have the ability to rapidly differentiate into antibody-secreting cells upon re-encounter of the pathogen. Their advantage is that they typically have a broader activity and are able to fine-tune their B cell receptor (BCR) specificity by their capability of re-entering germinal centres (GC). This allows memory B cells to recognize virus variants more efficiently [3,4,5]. Long-lived plasma cells and high-affinity memory B cells are typically generated in secondary lymphoid organs (SLO) during the GC reaction. This reaction critically relies on follicular helper T cells (Tfh), which drive B cell activation, proliferation, selection of high-affinity clones and differentiation into long-lived plasma cells or memory B cells [5,6,7]. Therefore, the ideal vaccine scheme not only relies on optimal B cell receptor epitopes but should also perfectly stimulate antigen-specific Tfh cells.

## 2. Generation of Tfh Cells in SLO

The spatially highly organized structure of SLO accommodates specific micro-anatomical environments that provide the necessary signals for Tfh cell differentiation [8]. T and B cell zones are organized by stromal cell subsets producing the chemokines CCL19/CCL21 or CXCL13, and by T and B cells expressing the corresponding chemokine receptors (CCR7 and PSGL-1 or CXCR5) [8]. During their generation from naive CD4^+^ T cells, Tfh cells migrate through different SLO compartments, which provide unique signals and drive the differentiation from pre-Tfh to GC Tfh cells.

Within the T cell zone, naive T cells are primed by contact with dendritic cells (DC) presenting their specific antigen [9,10,11]. At this early stage, fate commitment to the Tfh cell or alternative Th lineages, such as Th1, Th2, or Th17, is already determined. T cell fate critically depends on several cytokine pathways (Figure 1). For example, Tfh cell fate is facilitated by STAT3 [12] and inhibited by STAT5 signalling [13,14], while STAT1 signalling dictates a Th1 fate [15]. Hence, the cytokines IL-6 and IL-21, both connected to STAT3 signalling, promote Tfh differentiation [16,17,18], whereas IL-2-mediated STAT5 signalling inhibits upregulation of the lineage-defining transcription factor Bcl-6. Interestingly, Tfh cell precursors are themselves producers of IL-2 but do not express IL-2Rα [19,20]. In this context, IL-2 acts only in a paracrine fashion on neighbouring Th1 cell precursors, promoting the early fate of Tfh and Th1 cell development. In contrast, IL-21 as the major cytokine produced by Tfh cells, generates a positive feed-forward loop for Tfh cell generation [16,21]. The Tfh/Th1 fate decision is further reinforced by a direct and reciprocal inhibition of the two lineage-defining transcription factors, T-bet and Bcl-6 [22,23,24,25,26,27]. Therefore, modulation of the specific cytokine milieu might be a potential therapeutic way to boost or counteract early Tfh cell differentiation. In mouse models, neutralization of or supplementation with IL-2 successfully promoted or inhibited Tfh differentiation, respectively [13,14]. Moreover, the use of adjuvants that induce IL-6 production by DC could be a prospective blueprint to foster Tfh cell differentiation (see below).

Pre-Th1 cells upregulate their chemokine receptor CXCR3, which primes them for lymph node exit and migration into inflamed tissues [28]. In contrast, pre-Tfh cells concomitantly upregulate CXCR5 and downregulate CCR7, leading to their migration towards the border between T and B cell zone [29,30]. A specific cytokine milieu at the T/B border further promotes Tfh cell differentiation. This region is rich in a special subpopulation of IL-2R^high^ DCs, which act as a molecular sink for IL-2 [31]. In addition, these DC produce large amounts of IL-6 [32], which altogether provides the optimal environment for Tfh cell differentiation.

In parallel, cognate naive B cells are activated and migrate to the T/B border, facilitated by the reverse downregulation of CXCR5 and upregulation of CCR7 [33]. This strategical migration process enables the first interaction between antigen-specific T and B cells. Additionally, activated B cells, which highly express IL-2Rα, contribute to the low IL-2 levels at the T/B border and provide additional specific signals for further Tfh cell development, since mature Tfh cells are not generated in the absence of B cells [29]. Despite the fact that many of these processes are still not fully understood, one considerably acknowledged process is T cell co-stimulation via ICOS ([34], see below).

GC structures develop within the B cell zone and resemble highly organized microstructures in which T and B cell cooperation takes place. The positioning of GC is predefined by the presence of follicular dendritic cells (FDC), a stromal cell population producing CXCL13, which is the chemoattractant for CXCR5-expressing Tfh and B cells [35]. Additionally, this cell population has the unique property of capturing native antigens for a prolonged time to support the ongoing selection of B cells [36]. Only a fraction of Tfh cells develop into GC Tfh cells, which are characterized by the highest expression of CXCR5, PD-1, and IL-21 [29]. The remaining Tfh cells reside in the B cell zone, but only outside the GC structure or in interfollicular regions [37,38,39]. As previously mentioned, the particular signals that program a Tfh cell to evolve into GC Tfh cells are still inconclusive.

Within the GC, Tfh cells provide B cell help mainly through the production of IL-21 and CD40L, which are two major factors for B cell activation and differentiation [40]. Since the number of GC Tfh cells is much lower than the number of GC B cells, the availability of T cell help is the rate-limiting step for B cell selection. Only B cells with a high-affinity B cell receptors can efficiently capture antigens and present MHC class II/peptide complexes in high density, which in turn provides them with the best T cell help [41,42,43]. Dependent on received Tfh cell-mediated signals, GC B cells either undergo further rounds of somatic hypermutation to increase BCR affinity or leave the germinal centre to differentiate into long-lived antibody-producing PCs or memory B cells. B cells that lose antigen specificity undergo apoptosis [40]. Moreover, recent data proved a similar Tfh cell clonal selection based on TCR affinity and antigen availability [44]. After the termination of the GC reaction, memory Tfh cells can either remain in the lymphoid organ in which they have been generated [37,38,39,45] or turn into a circulating memory Tfh (cTfh) cell population (see below) (Figure 2).

Classical Tfh cells are generated in SLO, whereas in non-lymphoid tissues, Tph cells are typically the population providing help for the local differentiation of B cells into plasma cells. Memory Tfh, Tph and B cells can remain in SLO and non-lymphoid tissues. However, they are also found as circulating populations in the peripheral blood.

Apart from cytokines and chemokine receptors, costimulatory receptors are an additional attractive therapeutic target for modulating Tfh cell generation. Costimulatory signalling via CD28 is important for the very early upregulation of the transcription factor Bcl-6 in pre-Tfh cells [46,47], and the treatment of autoimmune patients with the CD28 co-stimulation inhibitor Abatacept results in reduced frequencies of Tfh cells [48]. Signalling via the inducible T cell co-stimulator ICOS is mandatory for the progression of pre-Tfh to fully differentiated Tfh cells [34,46,49,50]. Follicular mantle B cells exhibit high levels of ICOS-L, and intravital imaging studies have shown that close contact via ICOS/ICOS-L is required for full Tfh cell maturation [34,51]. Once a stable Tfh phenotype has developed, sustained signalling via ICOS is required to maintain the Tfh cell phenotype. Furthermore, experimental blockade of ICOS signalling results in a rapid upregulation of the transcription factor Klf2, subsequent downregulation of CXCR5 and other receptors important for Tfh cell localization in the B cell follicle, and ultimate cell relocation back to the T cell zone [46]. Upregulation of CXCR5 via ICOS is directly counteracted by signalling via PD-1, which contributes to the fine-positioning of T cells within the B cell follicle [52]. As another positive co-stimulator, signalling via OX40 has been shown to upregulate several Tfh-associated molecules [53].

## 3. Circulating Tfh Memory Cells in the Blood

Originally, Tfh cells were considered a terminally differentiated effector cell population, whose life span ends upon termination of the GC reaction. Although CXCR5^+^ CD4^+^ T cells in human peripheral blood were first described in 1994 [54], their relation to Tfh cells in SLO was not completely understood because T cells in the blood lack the lineage-defining transcription factor Bcl-6. This is a marked difference from Th1 or Th2 memory cells, which retain expression of T-bet and GATA3 also in the peripheral blood. Only in 2012, after using adoptive transfer mouse models, it was experimentally demonstrated that Tfh memory cells exist and can rapidly regain their original phenotype, including Bcl-6 expression, upon restimulation [37,55,56,57]. Despite this finding, other studies showed that mutations in ICOS [58], STAT3 [59], CD40L [60], or BTK [61] not only reduce the frequency of GC Tfh cells but also affect the number of cTfh cells initially suggested a common origin. Recent studies comparing tonsillar Tfh cells to donor-matched peripheral blood Tfh cells revealed clonal overlap and influenza-specific cTfh cell clones in both organs [62]. Consistently, analyses of phenotype, transcriptome, epigenetics, and TCR clonotypes of cTfh cells and Tfh cells from thoracic duct lymph support the hypothesis that Tfh cells exit LN into the blood via the thoracic duct [63]. Nevertheless, CXCR5^+^ T cells in blood are often referred to only as “Tfh-like” cells, whereby memory Tfh or cTfh cells describe their biological function much better.

Recently activated cTfh cells, which have just emigrated from the SLO, are characterized by high expression of ICOS and PD-1. Concomitantly, they also express higher levels of CD38, c-MAF and Ki-67 in combination with low CCR7 expression [55,56,64] and most strongly support B cell activation and differentiation in *in vitro* assays [55,56].

In addition to their activation status, cTfh cells have been further subdivided into three subsets according to the differential expression of chemokine receptors [65]. CXCR3^+^CCR6^−^ cTfh cells are referred to as cTfh1 cells and additionally express T-bet and secrete IFN-γ. CXCR3^−^CCR6^−^ cTfh2 cells express GATA3 and secrete IL-4, IL-5, IL-13. Moreover, cTfh2 cells are associated with class switch promotion to IgG and IgE. CXCR3^−^CCR6^+^ cTfh17 cells express RORγt and are distinguished by the cytokine secretion of IL-17A, IL-17F and IL-22, promoting the generation of IgG and IgA [65]. Using coculture assays with naive B cells, cTfh2 and cTfh17 were shown to efficiently stimulate B cell proliferation and antibody generation via IL-21 production [56,65], whereas this was not the case for cTfh1 cells [65,66]. However, cTfh1 cells can efficiently provide help for memory B cells [67,68,69]. Therefore, it is unclear whether this subset discrimination has relevance *in vivo*, as antigen-specific memory Tfh cells should always be accompanied by memory B cells. Moreover, in HCV-infected patients, CXCR3^+^ cTfh cells, but not their CXCR3^−^ counterparts, were shown to positively correlate with HCV-neutralizing antibody titers [69]. In addition, many IL-21/IFN-γ co-producing Tfh cells have been found in autoimmune and other inflammatory diseases [70,71,72,73].

Due to the limited access to human lymphoid tissue and the herein resulting difficulty to study ongoing immune responses in humans, most knowledge about Tfh cells has been obtained from mouse models. However, both resting and recently activated cTfh cells have been shown to be suitable biomarkers to detect infections, vaccination outcome success and the quality of memory responses. Elevated levels of activated cTfh cells positively correlate with protective antibody levels, plasmablasts, and memory cell formation [64,67,68,69,74,75,76]. Furthermore, increased cTfh cells are thought to reflect erroneous Tfh cell responses in autoimmune disorders, since a positive correlation has been observed with serum autoantibody titers, disease activity, and severity [55,77,78].

## 4. T and B Cells in Inflamed Tissues

T and B cells are not only found in SLO but also in many inflamed non-lymphoid tissues (Figure 2). Strong immune-activating conditions, such as influenza infection of the lung, promote the development of so-called ectopic lymphoid structures (ELS) that include T and B cell infiltrates and are also referred to as tertiary lymphoid tissue or inducible bronchus-associated lymphoid tissues (iBALT) in the lung. ELS functionally and structurally fully resemble their counterparts in SLO, which includes separated T and B cell zones, an FDC network, and the presence of classical Tfh- and GC B cells. In mice lacking all SLO, iBALT structures were sufficient to generate high levels of antibodies [79].

However, not all lung pathogens induce iBALT; for example modified vaccinia virus does, but this is not the case for *P. aeruginosa* [80]. Additionally, only few B cells and no FDC-containing ELS were detected in the lungs of severe COVID-19 patients [81,82,83,84]. The same applies to autoimmune and other chronic inflammatory diseases [85]. However, active T/B cooperation can also take place in FDC-negative disordered infiltrates, resulting in the local generation of GC-like B cells and antibody-producing plasma blasts [86]. How this is particularly organized, e.g., which cell types take over the function of FDC for antigen presentation, is still not clear. These responses are driven by tissue-resident Tfh-like cells that are characterized by high expression of PD-1 but lack of CXCR5 and Bcl-6. They provide potent B cell help due to their high expression of CD40L and production of IL-21 [86,87] and often express chemokine receptors for homing into inflamed tissues such as CCR2 and CX3CR1 [88,89].

These Tfh-like cells, also recently termed peripheral T helper cells (Tph), have now been identified in a variety of inflamed non-lymphoid human tissues, such as the joints of rheumatoid arthritis patients [88], the skin in systemic sclerosis [90,91] and pemphigus [92], the kidney in lupus nephritis [89], intestinal tissue in celiac disease [91], nasal polyps [93], salivary glands in Sjögren’s syndrome [94,95], the lung of sarcoidosis patients [96] and peritumoral tissue of breast cancer patients [97]. Although their presence in inflamed non-lymphoid tissues is a key feature of Tph cells, similar to Tfh cells, they can also be found as a circulating memory population in the peripheral blood, which seems to reflect an ongoing immune response in the inflamed tissue [88,91,94,95,96].

T and B cells in non-lymphoid tissues should not only be considered as a pathogenic cell population in autoimmune diseases. In barrier organs, such as lung or gut, they also provide a very efficient first line of defence against invading pathogens. Tissue-resident memory T (Trm) and B (Brm) cells have now been defined as an independent non-recirculating lineage, best characterized by their permanent expression of CD69, which antagonizes the lymphocyte exit receptor S1PR1 [98]. Although non-lymphoid tissues are typically dominated by CD8^+^ Trm, viral and non-viral infections in barrier organs also induce CD4^+^ Trm, which could be important for B cell help. This has been shown for mouse infection models [99,100] as well as human tissues [101,102,103]. A well-studied mouse model is influenza infection. This strong viral infection induces long-lasting iBALT structures containing Tfh as well as B cells within the lung tissue [104]. T and B cells can survive as tissue-resident memory cells for prolonged times and provide efficient protection from re-infection [105,106,107,108,109,110,111].Upon re-infection, lung-resident memory B cells rapidly develop into antibody-producing plasmablasts [109,111]. They also have a broader B cell receptor repertoire than B cells in the lung-draining lymph nodes and are therefore superior in generating neutralizing antibodies against viral escape mutants [4]. Similar observations were made for other airborne pathogens like *M. tuberculosis*, a mouse adapted SARS coronavirus, and *S. pneumoniae* [112,113,114].

However, lung-resident memory T and B cells are not a unique feature for severe viral infections and do not require the development of iBALT. In addition, in a protein immunization-based interstitial lung inflammation mouse model, tissue-resident memory Tph and B cells can be generated. Importantly, they are highly enriched in IgA^+^ memory cells, which is the exact isotype required for the effective protection of mucosal surfaces [86]. In T cell-specific *Bcl6* knock-out mice, which completely lack Tfh cells, protective neutralizing antibody levels can also be generated in an influenza vaccination setting [115]. Although it is not clear yet whether memory Tph cells are equally effective in driving memory B cell re-activation as classical Tfh cells, this is an important notion for the development of vaccination strategies.

## 5. Vaccination Strategies to Promote Tfh Cell Development

The production of Tfh cells can be manipulated at various levels, including cytokine milieus and costimulatory receptors. A rather simple and already widely used strategy to enhance antibody responses is the use of adjuvants in vaccination. Adjuvants act in two different ways: first, they contain specific immunostimulators to activate the innate immune system. These immunostimulatory ligands mimic pathogen- or damage-associated molecular patterns and include, for example, Toll-like receptor (TLR) agonists or C-type lectin receptor (CLR) agonists. These agonists stimulate DCs which subsequently enhance their antigen-presentation capacity [116,117,118]. At the same time, a proinflammatory cytokine milieu is generated, including cytokines such as IL-6, which foster Tfh cell development [32,119,120,121]. Second, many adjuvants exhibit a depot function for the immunogen. This is important since—unlike infectious agents that replicate in the body—vaccines are only bioavailable for rather a short time. This time can be prolonged by an adjuvant that slowly, but steadily releases antigens since the maintenance of Tfh cells is highly dependent on prolonged antigen availability [37,122]. Repeated injections of peptides have been able to overcome the normally limited antigen presentation by DCs and resulted in higher frequencies of Tfh cells [123]. A similar effect can be achieved with adjuvants, which have a very good depot function. In this situation, water in oil emulsions, which are more stable than oil in water emulsions, usually provide the best results [124,125]. Micro-osmotic pumps, which are implanted under the skin and constantly release their substance, also guarantee prolonged antigen availability and result in more than 10-fold increased antibody levels compared to a single bolus injection [126].

However, adjuvant use in humans remains very limited, mainly due to safety concerns [127,128,129,130]. Table 1 lists adjuvants that are currently in use, being tested in clinical trials, or of research interest, and describes their influence on Tfh cells. The most commonly used adjuvant in humans is still alum, and this adjuvant was used in vaccinations as early as the beginning of the 20th century [131]. Only within recent years, modern and more complex formulations such as MF59, the adjuvant systems (AS) AS01, AS03, AS04, and cytosine phosphoguanosine (CpG) have been approved for human vaccination.

The adjuvant alum induces robust CD4^+^ T cell and antibody responses [169], the effect of which has mostly been attributed to a depot effect, but which appears not to be required for adjuvant efficiency [170]. Besides this fact, alum strongly activates different pathways of the innate immune system (reviewed in [171,172,173]). An alum-containing adjuvant system is AS04, which consists of the TLR4 agonist 3-O-desacyl-4′-monophosphoryl lipid A (MPL) adsorbed to alum salts. The MPL adjuvant prolongs innate and humoral immune responses [135,136]. The adjuvants MF59 and AS03 are oil-in-water emulsion-based adjuvants and show improved tolerance compared to water-in-oil emulsions [149,174]. MF59 shows improved antigen uptake at injection sites [175] and induces strong antibody and T cell responses [176]. At the same time, AS03 is based on MF59 with the addition of DL-α-tocopherol, a bio-available form of vitamin E, promoting strong T and B cell responses [177]. CpG-ODN is a TLR agonist-based adjuvant that supports immunostimulatory effects of antigens and increases antigen-presenting cell activation [178,179]. Lastly, the liposome-based adjuvant system AS01 contains MPL as additional TLR4 agonist and elicits strong cellular and humoral immune responses [180,181].

In addition to the importance of using adjuvants, recent studies show an increase in Tfh and B cell responses by targeting Tfh-cell specific molecules. The addition of Fc-fused IL-7 to an influenza vaccine not only promoted Tfh and B cell responses in mice and cynomolgus monkeys, but also reduced the dose required for the generation of protective antibody levels [182]. The administration of a recombinant rabies virus expressing OX-40L promotes Tfh cells, GC B cells, and plasma cells and thus the protective antibody response in mice proposing OX-40L as a novel adjuvant strategy [183]. Besides Tfh cells, recent studies have demonstrated that the targeting of antigen-presenting cells is also beneficial. Antigen-fused monoclonal antibodies against DEC205, a surface receptor on a subset of DC, induced Tfh cell differentiation and strong B cell responses in mice [184]. HIV-1-vaccinated humanized mice showed enhanced B cell and memory Tfh cell responses after CD40-targeted boost vaccination [185], and targeting of antigen to MHCII molecules in mice significantly increased DC- and B cell-mediated presentation of p:MHCII, resulting in enhanced GC responses [186].

Another key aspect is the delivery of the vaccine. A study of malaria vaccination in non-human primate (NHP) models showed improved antigen-specific T and B cell responses in poly(lactic-co-glycolic acid)- (PLGA)-based synthetic vaccine particles (SVPs), co-administered with TLR-based adjuvants, compared to antigens delivered by liposomal formulations. One benefit of SVPs is their slow degradation and associated antigen retention in the lymph node [187]. In another study, nanoparticles of multilamellar lipids carrying the VMP001 antigen of the malaria-causing *Plasmodium vivax* promoted a broad humoral immune response in mice compared to vaccinations with soluble antigen or alum. Once more, nanoparticles showed improved antigen retention in draining lymph nodes [188].

A comparative study of three vaccines against HIV, zika virus and influenza virus with m1Ψ-modified mRNA lipid nanoparticles (m1Ψ-mRNA-LNPs) revealed robust and enhanced induction of Tfh cells, GC B cells, and neutralizing antibody responses compared to inactivated virus and adjuvanted protein vaccines in mice and NHPs. This may be due to sustained antigen presentation and additional adjuvant effects by mRNA-LNPs themselves [189]. These findings were confirmed in a recent study, comparing a SARS-CoV2-mRNA vaccine encoding the receptor-binding domain (RBD) and full-length spike protein of SARS-CoV2 with recombinant RBD protein (rRBD), formulated with the conventional adjuvant AddaVax. The mRNA vaccine showed superior ability to induce Tfh cell induction, GC B cell responses, and higher levels of neutralizing antibodies, likely due to prolonged antigen presentation [190].

Consideration of time delay and dosage of vaccination is equally critical. In NHPs, longer intervals between prime and booster immunization lead to increased GC B cell frequencies and neutralizing antibody titres [191]. The use of a continuous immunization strategy, for example via osmotic pumps, showed substantial advantages over conventional bolus immunization in both NHP and mouse models [126,191,192]. The positive effect of delayed fractional dose boosting on immune responses was confirmed in human studies with a malaria vaccine [193].

Given the fact that pathogens enter the organism via barrier organs, the generation of tissue-resident memory cell populations as a first line of defence should be an important strategy for the development of vaccination regimes. Although systemically or subcutaneously administered immunizations were shown to induce tissue-specific protection [194,195], local immunization strategies such as intranasal, intravaginal, or oral antigen administration appeared to be beneficial.

Intranasal immunization strategies elicited potent humoral immune responses in the murine lung, including high antigen-specific IgA levels and memory B cells [196,197]. Moreover, intratracheal administration of a polysaccharide-adjuvanted vaccine showed improved protection against aerosol *M. tuberculosis* infection in mice compared to parental administration. This could be attributed to the successful induction of lung-resident memory T cells [198]. Another study demonstrated the generation of lung-resident CD4^+^ memory T cells and protection against tuberculosis infection upon intranasal immunization with a recombinant influenza A viruses (rIAV) vaccine expressing *M. tuberculosis* peptides [199]. Moreover, supportive effects were seen in intravaginal administration [200,201]. In addition to effective memory T cell generation, orally administered inactivated enterotoxigenic *E. coli* vaccine in humans showed significantly increased levels of activated cTfh cells, which not only secreted increased levels of IL-21 but also expressed integrin ß7, indicating gut homing potential. Additionally, intestine-derived memory IgA responses were detected [202].

However, mucosa-based vaccination strategies also have some disadvantages. For safety reasons, the use of subunit vaccines is preferred. Yet, these require larger amounts of antigens to, for example, compensate for degradation in the gastrointestinal tract. Besides, there is need for adjuvants and delivery systems that allow local uptake by antigen-presenting cells [203]. This can be overcome by adapting the mode of delivery. Combined delivery of antigen and adjuvant on the same nanoparticle demonstrated increased frequencies of antigen-specific CD8^+^ tissue-resident T cells in the lung [204] and liposome-based vaccines proved to be more easily absorbed via the lungs, thereby activating alveolar macrophages [205].

Given the known importance of prolonged antigen availability for the effective generation of Tfh (see above) and in general memory cells, many strategies aim to improve antigen persistence. Intranasal administration of a biocompatible polyanhydride nanoparticle-based IAV vaccine (IAV-nanovax) generated effective humoral immune responses and CD4^+^ and CD8^+^ memory T cells via a long-lasting release of the antigen by surface erosion [206]. Recently, usage of alum adjuvant proved sustained antigen persistence as crucial for optimal tissue-resident memory cell induction in the lung [207], and another study unravelled the requirement of high amounts of antigens for the efficient generation of memory [208]. To prolong vaccine persistence, pH-responsive polymers were successful to increase retention in pulmonary antigen-presenting cells [204]. Another strategy is the stimulation of cytokines that specifically promote tissue-resident T cells [209,210]. Intranasal immunization with distinct TLR agonists coated pathogen-like particles generated phenotypically and functionally distinct populations of effector and memory T cells in lungs and airways of mice [211].

Dietary aspects can also have a huge impact on vaccination. Micronutrients such as zinc, copper, iron, and vitamins A, C, and D support immune responses [212]. Therefore, the WHO recommends combining the measles vaccination with a vitamin A supplementation in high-risk areas [213,214]. In a mouse model, vitamin A signalling was shown to directly promote the Tfh cell development via retinoic acid receptor α [215]. Moreover, retinoic acid as an adjuvant was shown to induce CCR9 and α4β7 expression on antigen specific CD8^+^ T cells, thereby promoting migration to intestinal mucosal tissues. As a micronutrient, selenium is particularly important for Tfh cell survival [216]. Glutathion peroxidase 4 (GPX4) is a selenium-dependent enzyme and prevents Tfh cells from ferroptosis. Selenium supplementation of mice increased Tfh cell numbers and promoted antibody responses after influenza vaccination. Furthermore, the metabolic hormone leptin has an important function for successful vaccine responses [217]. Leptin activates STAT3 and mTOR signalling pathways, thereby promoting Tfh cell differentiation. Fasting-induced leptin deficiency reduced the protective effects of influenza vaccination in mice and could be specifically reversed by leptin supplementation.

## 6. Conclusions

Tfh cells are a key determinant for the generation of high-affinity memory B cells and plasma cells. Their successful induction should be equally considered in the development of vaccination strategies. Tfh generation is highly dependent on specific cytokine milieus, and the use of appropriate adjuvants might be a way to selectively regulate their numbers. In addition to classical Tfh cells, Tph cells are a novel CD4^+^ T cell population that provides B cell help in non-lymphoid tissues. Tfh, Tph, and memory B cells all exist as lymphoid tissue-resident, non-lymphoid tissue-resident and circulating populations. At the same time, they have complementary roles in providing systemic and local protection. Tissue-resident Tph and memory B cells can play an important role as the first line of specific defence against pathogens in barrier organs such as the lungs. Therefore, it will be important to better understand the biology of tissue-resident memory T and B cells to find ways to specifically induce them for vaccination purposes.

## Figures and Tables

**Figure 1 vaccines-09-01074-f001:**
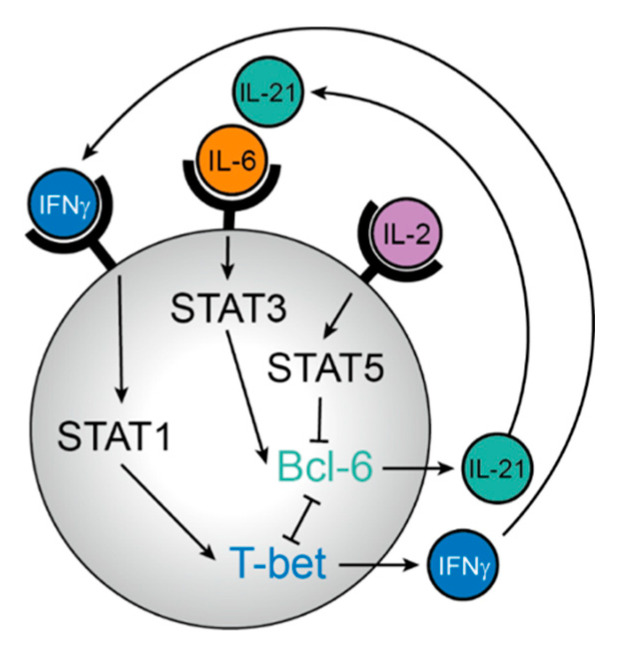
Regulation of T follicular helper (Tfh) versus T helper-1 (Th1) differentiation by cytokine signals. Induction of the Tfh cell master transcription factor Bcl-6 critically relies on binding of the transcription factor STAT3 (induced via interleukin- (IL-) 6 and IL-21) to its promoter and is inhibited by STAT5 (induced via IL-2). STAT5 does not only inhibit STAT3 binding to the *Bcl6* locus, but additionally induces the transcription factor Blimp-1, a direct antagonist of Bcl-6, thus promoting the development of non-Tfh effector T cells such as Th1 cells. As an additional feed-forward loop, the Th1 master transcription factor T-bet forms a repressive complex with Bcl-6 and inhibits a number of genes important for further Tfh cell development.

**Figure 2 vaccines-09-01074-f002:**
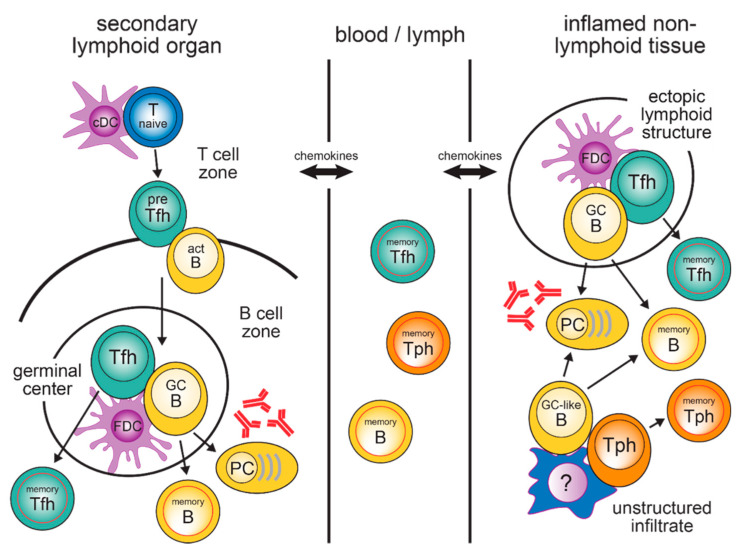
T cell/B cell cooperation in secondary lymphoid organs and inflamed non-lymphoid tissues. Abbreviations: act B, activated B cell; cDC, conventional dendritic cell; FDC, follicular dendritic cell; GC B, germinal centre B cell; PC, plasma cell; pre-Tfh, precursor Tfh cell; Tfh, follicular helper T cell; Tph, peripheral helper T cell.

**Table 1 vaccines-09-01074-t001:** Adjuvants and their influence on Tfh cells.

Adjuvant					Influence on Humoral Immune Response	Licensed in Humans	Literature
alum		alum			Tfh cell formation but lower numbers compared to combinations with TLR agonists	licensed	[128,132,133]
		AS04 (MPL + alum)			in combination with MPL strong activation of T and B cells, persistent antibody, and cellular responses, induces biased Th1 immune responses → targeting of viral infections	licensed	[127,134,135,136]
oil-in-water emulsions	squalene-based	MF59			promotion of potent immune responses in mice and humans	licensed	[137,138]
		AS03 (resembles MF59, combined with vitamin E)			potent Tfh cell activation in mice, high proportion of high-avidity antibodies after antigen recall in humans, persistence of cellular and humoral responses, induces marked antibody response, used in vaccines where antibody mediated protection is important	licensed	[127,139,140]
		GLA-SE (GLA + squalene)			promotes Tfh cell expansion in mice and men and effective, long-lived antibody production in humans, stable emulsion superior in enhancing adjuvanticity in GLA	clinical trials	[128,141,142]
		SLA-SE (SLA + squalene)			strong antibody and CD4^+^ T cell responses in mice		[143,144]
	saponin-based	AS02 (MPL + QS-21)			induces strong humoral and T cell mediated immune responses, used for pathogens that require strong T cell response, enhances humoral immune response in elderly people	clinical trials	[127,145]
Liposomes		AS01 (MPL + Saponin)			persistence of cellular and humoral responses in humans, robust innate stimulation, highly potent stimulation of CD4^+^ T cells and specific antibody responses in humans, designed to strengthen CD8^+^ T cell response	licensed	[127,140,146,147]
		AS015 (AS01 combined with CpG)			CpG promotes Tfh cell and antibody responses to influenza vaccination	not licensed	[148]
Water-in-oil emulsions		IFA			strong Tfh cell polarization in mice, strong side effects in humans and therefore not used, addition of CpG improved Tfh cell differentiation	not licensed	[124,149,150]
		CFA			triggers Tfh cells and additionally Th1, Th17 and Th2 responses in mice	not licensed	[124]
		Montanide			strong Tfh cell polarization in mice, side effects in humans	not licensed	[124,151]
immunostimulators		TLR4 agonists	LPS		not usable for humans due to toxicity		[152]
			synthetic TLR4 agonists	miltefosine (TLR4 and TLR9 agonist)	enhances Tfh cell responses and GC reaction in mice, induces both Th1 and Th2 antigen-specific cytokine responses, MTF improves efficacy of influenza vaccine against homologous and heterologous viruses by improving Tfh and antibody response, might be used for other than influenza vaccines as well		[153,154]
				MPL	included in AS01 and AS04		
				GLA	GLA-SE		
				SLA	SLA-SE		
		TLR7 agonists		3M052	promotes DC maturation and cellular response, enhanced Tfh cell generation compared to alum in NHP		[152,155,156]
		TLR9 agonists		CpG	included in AS015, combination of SARS-CoV-2 spike protein with CpG 1018 and alum elicited Th1-dominant immune responses with high neutralizing antibodies in mice		[157]
				IC31	induction of strong Th1 response in mice and humans, increase in Tfh cells in mice	clinical trials	[158,159]
		C-type lectin agonist	mincle agonist	CAF01	promotes GC responses and prolonged humoral responses in murine neonates, strong Th1/Th17 responses in mice	clinical trials	[158,160]
		STING agonist		chemically modified cyclic dinucleotide	induces Tfh and Th1 cell responses in neonatal cord blood, three-dose vaccination schedule is beneficial in mice leading to higher antibody titres; presence of cGAMP within HIV-derived virus-like particles enhanced adaptive immune responses, increased Tfh cell numbers in draining lymphnode		[161,162]
		Activin A			Study in NHP, HIV model: Activin A administration shows no differences in Tfh cell numbers, but decreased number of Tfr cells → either by promotion of Tfh cells or inhibitory role on Tfr development, improved antibody response and PC development		[163]
		CTA1-DD			CTA1-DD adjuvant well documented in mice, long term plasma cell and memory B cell development Effective mucosal and systemic adjuvant Can bind complement on FDCs, thereby directly affects DC function, directly influences gene transcription in FDCs, greatly upregulates Cxcl13 gene expression → strongly promotes GC B cell and Tfh cell development in neonate mice		[164,165,166,167]
		Adenosine deaminase-1			HIV-1 envelope (env) DNA vaccine: co-immunization with plasmid-encoded adenosine deaminase-1 in the context of an HIV-1 env DNA vaccine increases draining lymphnode Tfh cell frequencies and increases env-binding antibody in the serum of vaccinated mice → no increase in Tfh cell numbers compared to other groups but enhanced Tfh effector functions (increased serum antibody levels)		[168]

Abbreviations: AS, adjuvant system; CAF, cationic adjuvant formulation; CFA, Complete Freund’s adjuvant; cGAMP, cyclic guanosine monophosphate–adenosine monophosphate; CTA1-DD, cholera toxin A1 subunit (CTA1) fused via its C-terminal end to a dimer of the Ig-binding D region (DD); GLA-SE, glucopyranosyl lipid adjuvant–stable emulsion; IFA, incomplete Freund’s adjuvant; LPS, lipopolysaccharide; MPL, monophosphoryl-lipid-A; MTF, miltefosine; QS-21, plant extract derived from *Quillaja saponaria*; SLA-SE, second-generation lipid adjuvant formulated in a stable emulsion; Tfr, follicular regulatory T cell; TLR, toll-like receptor.

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
