# Peer review of "T Cell/B Cell Interactions in the Establishment of Protective Immunity"

_vaccines, 2021, doi:10.3390/vaccines9101074_

Round 1

Reviewer 1 Report

Ritzau-Jost J and Hutloff A have submitted the review entitled “T cell/B cell interactions in the establishment of protective immunity”. In this manuscript, the authors addressed the role of T cells, particularly T follicular helper cells, in the generation of B cells responses against immunogens, both in the secondary lymphoid organs and inflamed tissues. In addition, the authors also emphasize the different vaccination strategies, which will enhance the humoral immunity via Tfh cells. In a nutshell, the authors have written the manuscript very well and in a balanced manner.

This reviewer has minor suggestions, which will improve the manuscript quality.

  Line 154: Please delete the repeated word “that”.

Line 170, 172: It should be “symbol gamma” not “y”.

Line 203: “ELT” is supposed to be “ELS”.

In some places, pathogen names are not in italics, please take care.

Line 253: This reference deserves citation https://doi.org/10.1016/j.tips.2017.06.002

Line 262 and 263: It would be great to provide the definitions where terminology is new, e.g., water in oil emulsions, micro-osmotic pumps.

Table legend deserves the abbreviations section.

Line 376-390: It would be great if the authors provide a short table with the same information.

Line 376-390: Vitamin D and other micronutrients role should be emphasized. https://doi.org/10.1038/s41430-021-00949-8

Author Response

Q1: Line 154: Please delete the repeated word “that”.

R1: The double word has been deleted.

Q2: Line 170, 172: It should be “symbol gamma” not “y”.

R2: This has been changed.

Q3: Line 203: “ELT” is supposed to be “ELS”.

R3: This has been changed.

Q4: In some places, pathogen names are not in italics, please take care.

R4: All pathogen names are now written in italics.

Q5: Line 253: This reference deserves citation https://doi.org/10.1016/j.tips.2017.06.002

R5: The reference has been added.

Q6: Line 262 and 263: It would be great to provide the definitions where terminology is new, e.g., water in oil emulsions, micro-osmotic pumps.

R6: We added some explanatory information.

Q7: Table legend deserves the abbreviations section.

R7: All abbreviations are now either explained or have been written in full.

Q8: Line 376-390: It would be great if the authors provide a short table with the same information.

R8: We don't think that the only three micronutrients discussed justify a table to summarize the information. A very detailed table with multiple micronutrients is available in the review article mentioned below. However, for our article we wanted to focus on substances for which a direct effect on Tfh cells has been shown on the molecular level.

Q9: Line 376-390: Vitamin D and other micronutrients role should be emphasized. https://doi.org/10.1038/s41430-021-00949-8

R9: We now mention also other micronutrients and cite this very good review article.

Reviewer 2 Report

This is a very informative and well organized piece of review article focussing primarily on Tfh, T cell: B cell interaction during GC reaction, and various transcription factors and chemokines involved during an inflammatory response. I have a minor concern:

1) Line 170, 172 it is IFN-y and Roryt. Please modify it to IFN-g and Rorgt.

2)Although the authors have mentioned the characteristics of Tfh (upregulation of CXCR5, Bcl6), can the authors mention a few characteristic markers of Tph found in nonlymphoid organs.

3) Line 392. Did the authors miss Tfh cells?

4)Since Bcl6  is a transcrition factor (gene), the author should italicize it.

5)Since the authors have referred CD4 and CD8 Trms in nonlymphoid organs of mice and humans, can they phenotypically characterize these populations for the readers sake. And how are they different from Tem and Tcm?

Author Response

Q1: Line 170, 172 it is IFN-y and Roryt. Please modify it to IFN-g and Rorgt.

R1: This has been changed.

Q2: Although the authors have mentioned the characteristics of Tfh (upregulation of CXCR5, Bcl6), can the authors mention a few characteristic markers of Tph found in nonlymphoid organs.

R2: We now mention as additional characteristics the high expression of PD-1 and chemokine receptors for homing into inflamed tissues (lines 208-211).

Q3: Line 392. Did the authors miss Tfh cells?

R3: This has been changed.

Q4: Since Bcl6  is a transcrition factor (gene), the author should italicize it.

R4: In all cases, where we refer to the gene and not to the protein, Bcl-6 has been replaced by Bcl6 in italics.

Q5: Since the authors have referred CD4 and CD8 Trms in nonlymphoid organs of mice and humans, can they phenotypically characterize these populations for the readers sake. And how are they different from Tem and Tcm?

R5: This is now explained in a little more detail (lines 223-226).